# Point-of-Care Thoracic Ultrasound in Children: New Advances in Pediatric Emergency Setting

**DOI:** 10.3390/diagnostics13101765

**Published:** 2023-05-17

**Authors:** Silvia Bloise, Alessia Marcellino, Mariateresa Sanseviero, Vanessa Martucci, Alessia Testa, Rita Leone, Emanuela Del Giudice, Beatrice Frasacco, Pietro Gizzone, Claudia Proietti Ciolli, Flavia Ventriglia, Riccardo Lubrano

**Affiliations:** UOC di Pediatria e Neonatologia Ospedale Santa Maria Goretti—Polo Pontino, Dipartimento Materno Infantile e di Scienze Urologiche, Sapienza Università di Roma, 00185 Roma, Italy

**Keywords:** point-of-care, thoracic, emergency, children

## Abstract

Point-of-care thoracic ultrasound at the patient’s bedside has increased significantly recently, especially in pediatric settings. Its low cost, rapidity, simplicity, and repeatability make it a practical examination to guide diagnosis and treatment choices, especially in pediatric emergency departments. The fields of application of this innovative imaging method are many and include primarily the study of lungs but also that of the heart, diaphragm, and vessels. This manuscript aims to describe the most important evidence for using thoracic ultrasound in the pediatric emergency setting.

## 1. Introduction

In recent years, point-of-care thoracic ultrasound (POCUS) has been gaining more and more acceptance in clinical practice [1,2,3]. Its low cost, availability, rapidity, and repeatability [4] make it a suitable tool, especially in pediatric emergency departments, where accurate diagnosis in a short time is critical to ensure the best outcome for the patient. Specifically, in the emergency context, this method has three points of strength.

Primarily, thoracic ultrasound has a very rapid learning curve; in fact, pediatricians are increasingly using it [5]. This could stimulate emergency physicians to acquire this skill to quickly arrive at more accurate diagnoses.

Second, it is an investigation free of ionizing radiation. This aspect is crucial in the pediatric age, considering the cancer risks associated with radiation exposure and the fact that children have a high radiosensitivity of tissues and a longer expected lifetime in which cancer may develop [6].

Thirdly, thoracic ultrasound, given its proven diagnostic sensitivity and accuracy in numerous pediatric diseases, could be helpful for a rapid and initial assessment of patients. This allows the identification of patients at high risk of a life-threatening condition, acts as a follow-up tool for some acute pathologies, and avoids overloading diagnostic services with numerous instrumental investigations.

In fact, in an exciting communication in Jama journal [7], the ultrasound is defined as the fifth pillar to be added to physical examination at the patient’s bedside to improve the diagnostic-therapeutic pathway.

In view of these advantages and its increasing use even in pediatric settings over the past decade, this paper aims to review the more recent evidence of point-of-care thoracic ultrasound in children and its main applications in emergency departments to encourage emergency physicians to familiarize themselves with this technique that can be extremely helpful in performing diagnosis, managing critical situations, and guiding procedures.

## 2. Examination Technique and Ultrasound Characteristics in Healthy Subjects

A high-frequency linear probe (15 MHz or higher) should be used due to its better resolution. B-mode images are generally sufficient but can be supplemented by an M-mode study (especially for the pneumothorax assessment and diaphragmatic motion quantification) and color Doppler. A systematic approach to lung exploration should be used, especially at the first examination. Each hemithorax is divided into three areas: anterior area, delimited by the parasternal and anterior axillary lines; lateral area, within the anterior and posterior axillary lines; and posterior area beyond the posterior axillary line. Every area is subdivided into upper and lower halves. The probe must be placed perpendicular to the thorax and moved perpendicularly and in a parallel direction. The anterolateral thorax can be studied with the patient supine, while the posterior thorax can be explored either with the patient seated or in lateral or prone decubitus. Both longitudinal and transverse scans will be used so as to study the entire thorax.

In healthy subjects, the first layers of the chest are subcutaneous tissue and muscles. Then, we can see costochondral and sternal cartilages that appear hypoechogenic. In the axial, we can see the pleural line, a continuous hyperechogenic line that moves back and forth with the breaths. This movement is also known as “lung sliding.” The chest wall interface with the normally aerated lung results in a strongly reflective surface and produces a characteristic reverberation artifact, the A-lines, horizontal hyperechogenic lines that repeat at regular intervals equal to the distance between the skin and the pleural line.

## 3. Applications of Point-of-Care Thoracic Ultrasound in Pediatric Emergency Department

The main field of application of thoracic ultrasound is the study of the lungs; however, recently, the use of thoracic ultrasound has shown promising results in the study of other districts as well, summarized in Figure 1.

## 4. Respiratory Diseases

The lung can be considered a real achievement of ultrasonography. In fact, the ultrasound image of the lung is composed mainly of artifacts, and in recent years efforts have been made to exploit these artifacts and analyze all possible images that are generated under different pathological conditions, that is, as the degree of lung aeration decreases. In a normal lung, we can only view the pleural line (represented by the interface between the parietal and visceral pleura). The movement of the pleura line with breath acts is defined as lung sliding. The subpleural space consists of a series of reverberation artifacts called “A-lines”, represented as curvilinear echogenic lines parallel to the pleural line and repeating at regular intervals in depth. The lung ultrasound has recently been shown to be a valuable tool in the diagnosis of many respiratory diseases in the pediatric age.

### 4.1. Bronchiolitis

Bronchiolitis is a frequent reason for access to the emergency department. The diagnosis of this condition is mainly clinical; however, the recourse to instrumental evaluation is not rare.

Recently, lung ultrasound has been shown to be useful in the management of patients with bronchiolitis [8,9,10,11]. The typical ultrasound findings in patients with bronchiolitis are subpleural consolidations, seen as hypoechogenic areas with irregular margins, and the presence of coalescent B-lines (three or more B-lines in each intercostal space) up to the “white lung” or focal presence of multiple B-lines (one or two B-lines in each intercostal space) and abnormalities of the pleural line (Figure 2). In addition, the use of lung ultrasound in the emergency department in a patient with bronchiolitis could be helpful to confirm the diagnostic suspect and to define the prognosis. Different reports have shown that the ultrasound features or the ultrasound scores have a positive correlation with clinical, length of hospitalization, the need for oxygen therapy, or the need for admission to the pediatric intensive care unit [12,13,14]. This aspect is crucial since there is a lack of prognostic factors capable of predicting the course of the disease. Furthermore, lung ultrasound can diagnose pneumonia in infants with bronchiolitis, thereby not exposing the child to ionizing radiation. Biagi et al. have shown the sensitivity and specificity of ultrasound for the diagnosis of pneumonia in infants with bronchiolitis at 100% and 83.9%, respectively, compared with that of a chest X-ray, which was 96 percent and 87.1 percent, respectively [15].

Table 1 summarizes the most recent studies investigating the role of lung ultrasound in the diagnosis and prognosis of bronchiolitis.

### 4.2. Pneumonia

In clinical practice, a chest X-ray is used as a first-choice exam in patients with suspected pneumonia. However, the current guidelines recommend chest radiography only in patients with severe clinical conditions or uncertain diagnoses [16,17]. Therefore, LUS could be a valuable tool to guide the diagnostic-therapeutic course and in the follow-up of these patients. In community-acquired pneumonia (CAP), lung ultrasonography shows typical features; consolidations appear as hypoechogenic areas with evidence of air bronchogram within and an ecostructural “liver pattern” (pulmonary hepatization or tissue-like sign) with clear margins or surrounded by confluent B-lines (Figure 3) [18]. This consolidative picture differs from that where interstitial involvement prevails (as in the case of viral infections or Mycoplasma), characterized instead by confluent B lines up to the appearance of the pulmonary white lung. Growing evidence shows that LUS has good sensitivity, specificity, and accuracy in diagnosing pneumonia compared with chest radiography [19,20,21,22,23,24,25]. Similar results are derived from different meta-analyses conducted on the adult population for which the comparison was made not only with radiography but also with chest-CT, considered the gold standard for respiratory pathology and which obviously cannot be used in newborns and children due to high exposure to ionizing radiation [26,27].

Furthermore, lung ultrasound allows us to differentiate between pneumonia and atelectasis. The difference lies in the appearance of the air bronchogram, defined as dynamic due to the movement of air inside in the case of pneumonia and static in the case of atelectasis [28]. The ecographic feature in atelectasis is that of pulmonary hepatization. In the affected area, the air artifacts are not visible (lines A absent), and the lung appears as a parenchymatous-like structure with high density. The normal pleural sliding is abolished, and the “lung pulse” can be highlighted, i.e., the pulsation of the pleural line synchronous with cardiac activity [29].

When atelectasis is secondary to a massive pleural effusion, the ultrasound image is called a “jellyfish sign,” in which the lung is seen floating in the pleural fluid [30].

In a recent prospective study conducted on 120 adult patients, through the evaluation of dynamic air bronchograms and color Doppler imaging to differentiate pneumonia and atelectasis, the ultrasound is shown as an accurate and directly bedside available tool to distinguish the two conditions [31].

In this context, Ullmann et al. have shown the utility of ultrasound in the diagnosis of atelectasis, especially in patients with neuromuscular disease who are often exposed to ionizing radiations. In these patients, the use of this non-invasive method could provide relevant information for clinicians and respiratory physiotherapists [32].

Ultimately, lung ultrasound has also been used in children with COVID-19. Although SARS-CoV-2 infection has a milder course in children than in adults [33,34,35], recent studies have also shown the presence of ultrasound abnormalities during the disease in this age group. The main ultrasound findings of SARS-CoV-2 infection in children include pleural line irregularities, subpleural consolidations, and vertical reverberation artifacts and areas of the white lung [36,37]. Musolino et al. also described two cases of spontaneous pneumothorax, the first ones reported in the literature in adolescents with SARS-CoV-2 infection, diagnosed by lung ultrasound [38].

No study has demonstrated a correlation between ultrasound pictures and disease severity. However, ultrasonography could help detect COVID-19 pneumonia, select the most at-risk patients with extensive lung involvement and refer them to tertiary care centers, and manage patients with a negative nasal swab where ultrasound findings are suggestive of disease through precautionary isolation of the patient and repetition of another swab [39].

In this context, lung ultrasonography could also help in monitoring any sequelae of SARS-CoV-2 infection in children. Denina et al. followed up for 4 months with 25 children previously hospitalized for COVID-19. At the follow-up, they performed a lung ultrasound observing a mild interstitial pattern in three patients and multiple subpleural consolidations in the other two cases [40].

Table 2 summarizes the most important studies investigating the role of lung ultrasound in the diagnosis of pneumonia in children.

### 4.3. Lung Abscess

The diagnosis of lung abscess often requires more extensive instrumental investigations than chest X-rays. Recently, lung ultrasound has demonstrated its role in the diagnosis of this condition. On ultrasound, the abscess is seen as a well-demarcated capsular structure surrounding a hypoechoic core non-vascularized and follows its evolution over time [41] (Figure 4). In this way, additional investigations, such as computed tomography (CT), can be avoided with significant radiation protection benefits, especially in the pediatric age group. Furthermore, through convex-probe endobronchial ultrasound, it is possible to guide procedures, such as transbronchial needle aspiration, and to help physicians with differential diagnosis and treatment choices [42].

### 4.4. Asthma

Even in asthmatic patients, the use of lung ultrasound has shown interesting results. The main ultrasound findings in children with asthma are B lines, pulmonary consolidations, and/or pleural abnormality. These artifacts can be seen both outside of acute exacerbations [43] and during exacerbations. In particular, Dankoff et al. showed that pathologic lung ultrasound scans performed in asthmatic patients accessing the emergency department were related to increased antibiotic use, prolonged length of stay in the emergency department, and increased rate of hospitalization [44].

### 4.5. Pulmonary Edema

Another well-established application of lung ultrasound is in the diagnosis of pulmonary edema. This is characterized by the accumulation of fluid in alveoli and interstitial spaces following cardiogenic and noncardiogenic causes and frequently occurs after cardiac surgery. Its diagnosis is performed through visualization and quantification of B-lines. In the context of pulmonary edema, B-lines indicate an excess of extravascular lung water. They represent a consequence of the thickening of inter-alveolar septa and also alveolar flooding. We can describe two types of B-lines: B3 lines and B7 lines. Multiple B-lines that are 7 mm apart are caused by thickened interlobular septa characterizing interstitial edema. Instead, B-lines that are <3 mm apart are associated with ground-glass areas characterizing alveolar edema. Finally, when we observed B-lines greater in number and thicker, more severe pulmonary edema is expected: interstitial pulmonary edema with focal B-lines and alveolar pulmonary edema with confluent B-lines in a fully echogenic lung. In the context of pulmonary edema, the presence of B lines indicates an excess of extravascular lung water (EVLW). The gradual separation between B lines (B3 to B7 lines), reduction in the total number of B lines, and in cases with a favorable course, the disappearance of B lines, all indicate a reduction in EVLW and a successful response to treatment [45]. Different ultrasound scanning protocols and scores are described to evaluate pulmonary edema. The normal lung is black (no signal), the abnormal wet lung with interstitial pulmonary edema is black and white (with focal B-lines), and the lung with alveolar pulmonary edema is white (confluent B-lines in a fully echogenic lung) [46,47,48,49,50]. Lung ultrasonography could also allow differentiation between cardiogenic pulmonary edema and noncardiogenic alveolar interstitial syndrome. In a recent study, M-mode ultrasonography proved to be a useful tool to distinguish the two causes on the basis of pleural and subpleural morphologic features. In particular, a fragmented pleural line and a vertical subpleural pattern, the existence of spared areas of normal parenchyma (no homogenous distribution of B lines), and reduced pleural motion are associated with patients who have a noncardiogenic alveolar interstitial syndrome. In contrast, a continuous pleural line and a vertical subpleural pattern are associated with patients who have cardiogenic pulmonary edema [51].

### 4.6. Pleural Pathologies and Complications of Pneumonia

In addition, lung ultrasound has shown its efficacy, especially in pleural pathologies and complications of pneumonia. Primarily, lung ultrasound showed a sensitivity and specificity close to 100% in the diagnosis of pleural effusion, with an accuracy major than the chest X-ray. The correct way to begin an ultrasound to evaluate pleural effusion is to start from the diaphragm. At this point, it is essential to identify five structure keys: liver/spleen, diaphragm, chest wall, lung, and effusion. Pleural effusion can be visualized by ultrasound as a dark and anechogenic region located above the diaphragm that also determines the disappearance of the mirror image of the liver and spleen in the lung fields, which can be visualized in the healthy lung (Figure 5). Furthermore, large pleural effusion, compressing the lung, can take on ultrasound characteristics such as liver parenchyma. The chest X-ray is a diagnostic technique used to detect pleural effusion. In the anterior-posterior view, effusions >200 mL can be recognized. This means that up to 10% of payments are not recognized. The same limitations apply to chest CT [52].

Instead, lung ultrasound allows for calculating the amount of liquid and identifying pleural effusions smaller than 15 mm [53,54,55,56]. A minimum volume of 20 mL is more easily detectable, while there is a sensitivity of 100% for effusion >100 mL. In the differential diagnosis, the grayscale US may not be adequate to distinguish the pleural effusion compared to a thickening. For this reason, color Doppler imaging can be used, which instead has a specificity of 100%. Despite having a lower sensitivity, this technique is more accurate on payments of lesser entities [57].

Another advantage of lung ultrasound is to distinguish the nature of the fluid in the effusion, differentiating between transudate and exudate. The transudates are anechoic, pleural effusions with septation or internal echogenicity are exudates, and the association of pleural thickness and lung changes also suggests exudates [58]. In this area, lung ultrasound can also play an important role in the operative decision of whether or not to drain an effusion, instill pharmacological agents into the pleural space, or proceed with more invasive surgery. Guidelines recommend the use of ultrasound during thoracentesis. This is because there has been a better outcome and fewer complications associated with the procedure [59]. Pneumothorax is the most common complication during this procedure. A study has shown how ultrasound can reduce this complication. Operator experience can also play an important role [60].

Second, lung ultrasound is useful in the diagnosis of pneumothorax (PNX). Pneumothorax occurs when air collects between the parietal and visceral pleurae, causing the lung parenchyma to collapse. The ultrasound features of pneumothorax in B-MODE are the absence of lung sliding (the pleural interface remains immobile despite respiratory acts) followed by a stretch in which normal pleural sliding is highlighted and lung point (the fleeting appearance of a lung pattern lung sliding or pathologic comet-tail artifacts, replacing a pneumothorax pattern, absent lung sliding plus exclusive horizontal lines). The lung point is the representation of the transition point between PNX and normal lung [61]. In the M-mode are the stratosphere signs, also called bar code signs, and the equivalent of the lung point is given by the succession of a stretch of bar code signs interspersed with a stretch of seashore signs [62]. The search for these lung points, which can be performed at the patient’s bedside and repeated on a daily frequency, makes it possible to monitor the disease process while avoiding other investigations that are invasive and the problems arising from the subject’s travel to other departments of radiology. In the diagnosis of chest X-ray for pneumothorax, ultrasonography has also proved superior [63,64]. In a recent meta-analysis, including 13 studies and 1270 patients, the diagnostic accuracy of lung ultrasound performed by non-radiologist physicians in trauma patients was superior to supine chest X-ray, independent of the type of trauma, type of operator, or type of probe used. This is because the anteromedial and apical regions of the lung, where air tends to accumulate, are difficult to explore with X-rays [65].

For example, in mechanically ventilated infants, pneumothorax remains a critical condition that can lead to death if not treated promptly. The standard is a chest X-ray, but according to a study conducted by Cattarossi et al., ultrasound can still play an equal role compared to X-ray. The anatomy of the newborn facilitates the use of ultrasound. Another feature that makes it an interesting method is the speed of execution. While it may take some time to request and perform an X-ray, the ultrasound can easily be performed at the patient’s bedside, allowing for rapid diagnosis and resolution of the problem [66].

Third, lung ultrasound could be useful in the diagnosis of pleural empyema. It is defined as pus in the thoracic cavity due to pleural space infection and has a multifactorial underlying cause, although the majority of cases are post-bacterial pneumonia [67]. On ultrasonography, it is seen as a hypoechoic lesion with complex-septated effusions, passive atelectasis, width uniformity, and smooth luminal and outer margins. In addition, through color Doppler, we can differentiate between empyema or peripheral abscess [68,69,70]. Color Doppler ultrasound vessel signals in pericavitary consolidation are a predictor of lung abscess. This is because empyema usually involves the compression of the lung parenchyma, resulting in reduced blood flow [71].

It is essential to be able to distinguish empyema from parapneumonic effusion. This is because the type of treatment is different. In the first case, it is conservative with antibiotic therapy. Secondly, it can prevent drainage. Ultrasound can play a role in this. It has been seen that rx does not allow the distinction between the two pathologies. A chest CT scan may be helpful. The empyema appears as a lenticular lesion, with a smooth and well-differentiated margin compared to the normal lung, while the parapneumonic effusion has an irregular character, with no distinction between a healthy lung and a lesion. These same features can be used as a guide in ultrasound. The difference is given by identifying air-fluid levels, an element missing from the empyema [68,69,70].

Furthermore, in an interesting case report, the ultrasound has facilitated a rapid and accurate diagnosis of empyema necessitans [72], a rare complication of pleural empyema characterized by the dissection of pus through the soft tissues of the chest wall and eventually through the skin. It is very important to diagnose empyema necessitans and not confuse it with a superficial abscess, as the treatment choice is different. A superficial abscess is based on local drainage, while the empyema necessitans will require either chest tube drainage, open drainage, or even decortication in specific cases.

## 5. Thoracic Ultrasound and Other Applications

### 5.1. Acute Heart Diseases

Point-of-care ultrasound has recently shown promising results in studying heart diseases, allowing real-time visualization of the heart and diagnosing life-threatening conditions. For example, pediatric emergency physicians with targeted training can accurately diagnose pericardial effusions, cardiac contractility abnormalities, left ventricular dysfunction, cardiac tamponade, dilated cardiomyopathy, congenital heart disease, and infective endocarditis [73,74,75,76]. Thanks to all the valuable elements, it appears to be a useful technique for evaluating hemodynamic stability or instability in critically ill patients. The measurement of systolic function in critically ill pediatric patients seems to have played an important role in recent years. This is because, according to some studies, it seems to be able to become part of a multimodal approach, together with other elements such as clinical examination and invasive pressure monitoring in patients with septic shock [77,78]. In particular, this type of ultrasound has proved useful in various states of shock but also in cardiac arrest. It is possible to evaluate the state of preload during resuscitation [79]. Moreover, this technique allows for evaluating cardiac contractility in a short time, obtaining interpretable values within less than 10 s, thus reducing interruptions during resuscitation and ensuring better care [80]. Therefore, performing cardiac point-of-care ultrasound on a child with suspected cardiac disease can be life-saving, allowing optimal therapeutic strategies to be implemented [81].

Thoracic ultrasound also plays a central role in children undergoing cardiac surgery [82,83,84]. In these patients, it is essential to evaluate the degree of pulmonary congestion through the quantification of B-lines. Furthermore, pulmonary complications are frequent after cardiac surgery, including atelectasis, effusion, pneumonia, pneumothorax, diaphragmatic motion anomalies, and retrosternal clots. Lung ultrasound is exceptionally sensible in all these conditions and could be helpful for the follow-up of these patients, avoiding the exposition to ionizing radiation and guiding interventional procedures such as drainage insertion for pleural effusion and pneumothorax.

A recent application of cardiac point-of-care ultrasound in the context of COVID-19 disease detects multisystem inflammatory syndrome (MIS-C) findings. Del Monaco et al. [85] described the ultrasound findings in a case series of MIS-C patients admitted to the emergency department. They were cardiac hypokinesis inferior, vena cava dilatation, peritoneal fluid, and lung interstitial syndrome, suggesting that POCUS can be useful in the patient with suspected MIS-C for a primary assessment and for monitoring the patient at risk of worsening cardiovascular function.

### 5.2. Assessment of Volume Status and Conditions of Hypovolemia

Pediatric dehydration is a frequent reason for admission to the emergency department and is one of the most common reasons for hospitalization in children of all ages. The degree of dehydration can be difficult to quantify, and there are currently no sensitive or specific laboratory values or clinical scores to assess it accurately. In addition, the management of patient dehydration also varies greatly across centers. In this context, ultrasonography has been proposed as a non-invasive tool to assess and follow the status of dehydration [86,87]. The diameter of the inferior vena cava and the right ventricle have been proposed as possible ultrasound markers of a hypovolemic condition; in fact, several studies conducted on adults have shown that they are consistently low in hypovolemic subjects compared with euvolemic subjects [88,89]. This field of application of ultrasound may be useful in patients with moderate dehydration to identify the patient at risk early and set up adequate rehydration, especially in patients in shock. In children in shock, the situation is more complex. Indeed, the choice of optimal therapy, between fluids or inotropes, must be based on the signs of preload: jugular turgor, rales at the lung bases, and hepatomegaly [90]. In this context, of increasing interest is the use of ultrasonography in the assessment of preload responsiveness through the use of dynamic rather than static assessment parameters, such as changes in vena-caval or cardiac dimensions, which appear to be more predictive of volume responsiveness [91]. In these patients, it is crucial that resuscitation is guided by a reliable and accurate method of assessing volemic status to avoid conditions of hypovolemia or fluid overload, both of which are dangerous for the patient [92]. This is precisely why resuscitation with fluids is the first step in managing shock. It is essential for this to evaluate the response to fluid therapy using ultrasound. This evaluation can be performed by observing two aspects. The first is to evaluate the diameters of the vena cava, while the second is to analyze the respiratory change of the aortic peak flow rate during both exhalation and inspiration. It has been seen that the value of cardiac output, which varies during the states of sepsis and hypovolemic shock, can be quickly obtained. This means that compared to laboratory and clinical data alone, a better and faster approach can be obtained in the event of dynamic instability [93].

Furthermore, another valuable use of POCUS is as a guide in vascular accesses, especially in pediatric age, where finding venous access is often difficult and is associated with pain and crying for the child. In fact, different reports showed that the use of POCUS improves the rate of successful peripheral intravenous access [94], allowing the preservation of the venous heritage in cases of moderate severity to reduce stress for children. According to Vinograd et al., the use of ultrasound in venous accesses, in addition to being a technique with a higher success rate than the traditional one, seems to be linked to a lower complication [95].

### 5.3. Traumatic Diseases and Early Detection of Signs of Child Abuse

Ultrasound is an essential tool in managing the patient with polytrauma and is ideal in the primary evaluation of trauma patients because it can accurately reveal hemorrhagic effusion in body cavities (such as pericardial and pleural spaces) [96,97]. In patients with traumas, time is precious, and ultrasound, due to its rapidity, is the ideal examination to guide therapeutic choices. In recent years, an extension of the protocol, the Focused Assessment with Sonography for Trauma (FAST) protocol, E-FAST (Extended-FAST), has been developed, adding to the evaluation of the heart and abdomen that of the thoracic cavity [98]. The thoracic ultrasound is sensitive in the diagnosis of hemothorax, pneumothorax, pleural effusion, and chest injuries.

In particular, point-of-care ultrasonography seems to have a higher sensitivity in chest injuries than abdominal injuries [99].

Furthermore, recent reports have shown that the detection of thoracic bone fractures was higher with lung ultrasound than with chest radiography, especially for the diagnosis of rib fractures [100,101,102] (Figure 6). This aspect is crucial in the pediatric age; rib fractures have high specificity for child abuse and can be difficult to confirm on an initial radiographic skeletal survey. In addition to the evaluation of the ribs, thoracic ultrasound, compared to radiography, allows an evaluation of costal cartilagines, costochondral junctions, and non-displaced fractures. It also allows the evaluation of sternal fractures, which may indicate a significant traumatic mechanism with potential heart damage. Therefore, using thoracic ultrasound in the emergency department could help identify children at risk of suspected abuse early. In fact, the use of thoracic ultrasound could be effective in the early identification of rib fractures, thus confirming the suspicion of child abuse. The use of this radiological technique is also useful in decreasing the radiation dose to which children may be subjected with radiography [103].

This application is also useful in accidental traumatic injuries, especially given the possibility of rapid detection of any associated pneumothorax.

### 5.4. Assessment of Drowning Victims

Drowning has a high frequency in pediatric age, representing one of the most common injury-related causes of death in children one to four years of age. Pulmonary complications are the most common in drowning victims, including non-cardiogenic pulmonary edema, acute respiratory distress syndrome (ARDS), and pneumonia. Lung ultrasound can accurately diagnose these conditions. Currently, there are only two reports in the literature, one in adult patients and the other in a child, that describe the use of lung ultrasound in the evaluation of a victim of a near-drowning accident [104,105]. In the first case report, the lung ultrasound diagnosed non-cardiogenic pulmonary edema. In the second case report, pneumonia, without recourse to chest X-ray. Both these reports emphasize the clinical role of lung ultrasound for the initial assessment and follow-up of patient victims of drowning, especially during follow-up, where the risk of developing complicated pneumonia caused by water-borne pathogens, such as Aeromonas, Pseudomonas, and Proteus is very high. This allows the child’s lungs to be closely monitored without exposing them to ionizing radiation.

Therefore, we believe this method could be included in the emergency department’s diagnostic pathway of the child drowning victim.

### 5.5. Diaphragm Ultrasound

Point-of-care ultrasound can be used to evaluate the diaphragm. This application is already widespread in adult patient settings, particularly in critically ill patients. In these patients, the ultrasound proved to be an accurate tool for detecting diaphragmatic dysfunction, predicting extubation success or failure, monitoring respiratory workload, and assessing atrophy in mechanically ventilated patients [106]. This field of application is also making its way into the pediatric field. The diaphragm represents the most important respiratory muscle; different factors, such as mechanical ventilation, pulmonary and thoracic pathologies, and abdominal surgery, can result in reduced functioning, causing diaphragmatic dysfunction associated with respiratory insufficiency. The diaphragmatic parameters that can be evaluated are diaphragm thickness at the end of inspiration and expiration, thickening fraction, diaphragm excursion, inspiratory slope, expiratory slope, and total duration time of the respiratory cycle.

Primarily, the diaphragm point-of-care has been used in critically ill children to guide ventilator titration and weaning [107,108]. Recently, exciting applications of diaphragmatic POCUS have also been described in common diseases in the pediatric age, pneumonia and bronchiolitis. In particular, Sik et al. showed as some diaphragm parameters, especially thickening fraction and inspiratory and expiratory slope, can predict the severity of pneumonia, the need for respiratory support, and outcomes [109]. Moreover, in bronchiolitis, different studies showed that ultrasound diaphragm parameters (diaphragm excursion, inspiratory excursion, thickness at end-inspiration) are related to the clinical outcomes of the patients [110,111]. In particular, they seem to correlate with the length of stay, hours of oxygen delivered, and the need for respiratory support, helping evaluate an infant with bronchiolitis. A trained emergency pediatrician can easily perform this technique, which permits better monitoring and evaluation of respiratory function in infants with bronchiolitis, suggesting clinical management. POCUS can also be used to evaluate the lower thoracic and upper abdominal regions to exclude pathologies adjacent to the diaphragm. Other clinical conditions may result in abnormal contouring of the diaphragm, such as air trapping, status asthmaticus, and bronchopulmonary dysplasia. This field of application is critical in the pediatric emergency setting, enabling rapid selection of the most at-risk patients and setting up targeted treatment strategies.

### 5.6. Foreign Body Identification

Currently, the diagnostic test indicated for the identification of ingested foreign bodies is radiography. However, this test does not identify radiolucent foreign bodies. Ultrasound has been shown to have greater sensitivity than radiography in detecting nonradiopaque foreign bodies, such as objects made of wood, acrylic, and plastic [112,113]. These usually present as hyperechogenic images with posterior shadow cones. The application of ultrasonography, in this context, could be useful for several reasons; primarily for the rapid identification of the ingested foreign body; second, to identify foreign bodies not visualized on radiography; and finally, to guide and confirm their complete removal.

## 6. Limitations of the Exam

Although thoracic ultrasound has proven to be a valuable tool in the diagnosis, prognosis, and follow-up of a wide variety of acute pediatric conditions, it is necessary for operators to be aware of the inherent limitations of the technique. The most important limitation is that the exam is dependent on the operator; therefore, operator training and expertise are essential elements. Moreover, the time to perform the exam is also an important consideration; while the CXR takes only a couple of minutes, a complete lung US can take 20 min or even more, which is crucial in the emergency context. Finally, another important limitation is that the performance of lung ultrasound for central lesions is inadequate; in fact, the technique does not identify lung consolidations that do not reach the pleura. Fortunately, almost 98.5% of lung consolidations in adults reach the pleural line. Taking into consideration that a child’s pulmonary volume is smaller, the rate of pleural contact in pediatric pneumonia should be at least equal to that in adults. Furthermore, certain anatomical characteristics in children, such as a thinner chest wall and smaller thoracic width and lung mass, facilitate ultrasound imaging and ensure good-quality images of the lung.

## 7. Conclusions

The use of ultrasound has increased significantly in emergency medicine and intensive care. Thanks to technological advances with the miniaturization of electronics and increased accuracy, we may have an ultrasound probe that could become the “stethoscope of the future.” This method has proven to be sensitive and accurate in the diagnosis of major respiratory, cardiac, and traumatic diseases. In addition, the use of ultrasound has proven useful in many other pediatric emergency conditions, such as assessing dehydration status, identifying foreign bodies, evaluating the diaphragm, and finally finding venous accesses with the significant advantage of not using ionizing radiation. It is a focused examination performed and interpreted at the patient’s bedside, allowing real-time enrichment and completion of the objective examination. Its use for diagnostic and therapeutic purposes could improve health care in the pediatric emergency setting by a faster diagnosis of time-sensitive critical conditions, selecting patients who need more detailed evaluation, and minimizing delays in care and procedural complications. This approach can save billions of dollars annually across health systems, thereby contributing to better use of economic resources and improved quality of care.

## Figures and Tables

**Figure 1 diagnostics-13-01765-f001:**
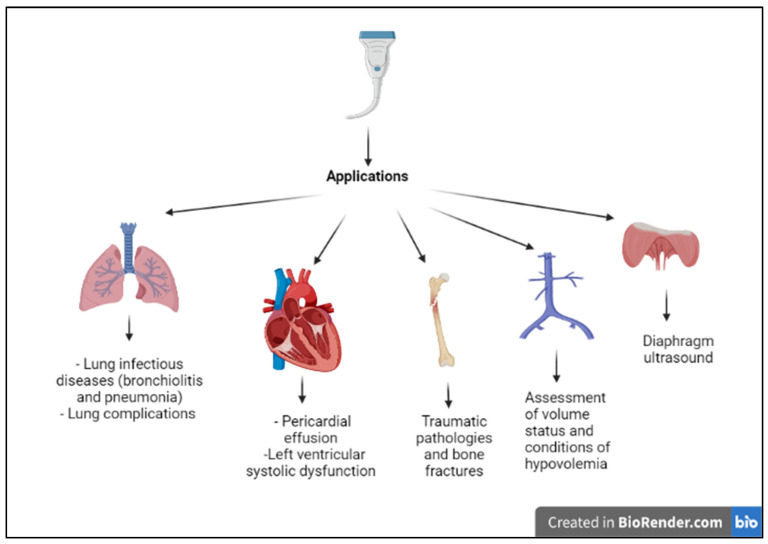
The figure shows the possible applications of the point-of-care thoracic ultrasound in the Pediatric Emergency Department: the main application is in respiratory diseases, followed by acute heart diseases, traumas, assessment of volume state and hypovolemia, and assessment of the diaphragm. The figure was created in BioRender.com (https://www.biorender.com/, accessed on 1 August 2022).

**Figure 2 diagnostics-13-01765-f002:**
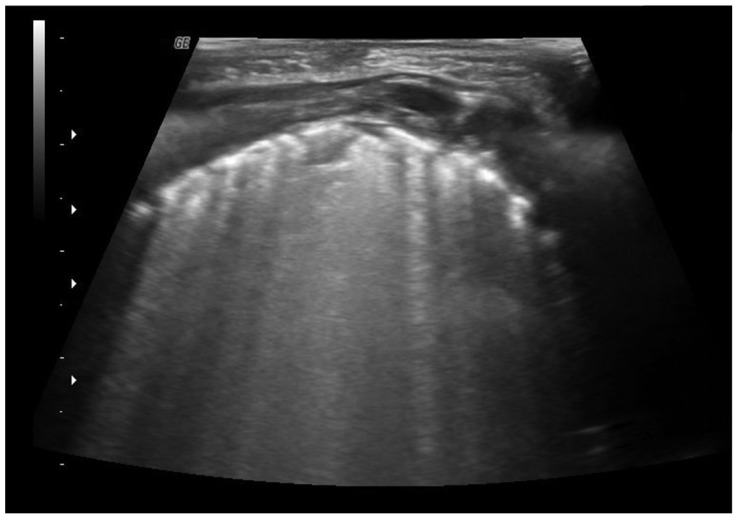
The figure shows coalescent B-lines, irregularity of the pleural line, and subpleural consolidation.

**Figure 3 diagnostics-13-01765-f003:**
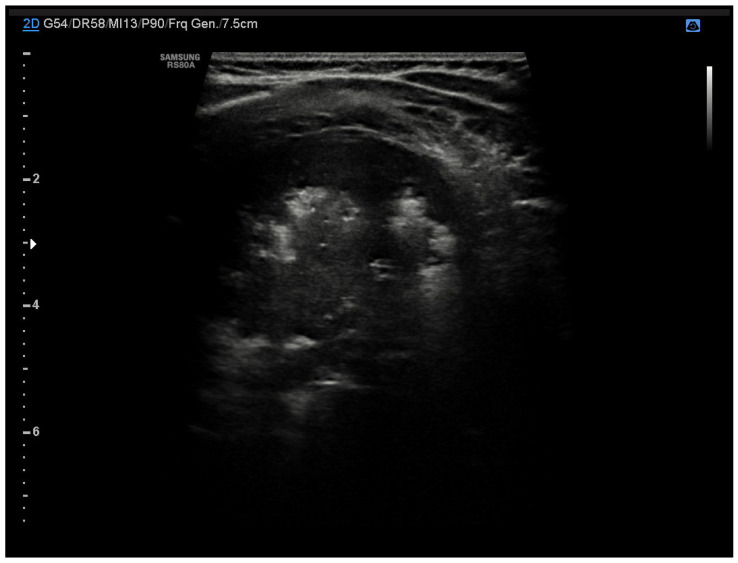
Pneumonia. The figure shows a consolidation, hypoechogenic area with evidence of air bronchogram within and an ecostructural “liver pattern”.

**Figure 4 diagnostics-13-01765-f004:**
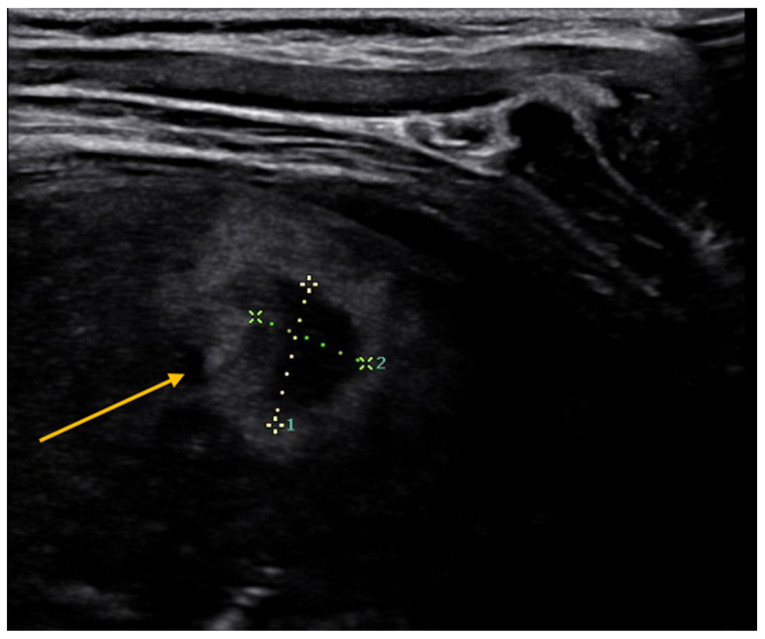
Lung abscess. It is seen as a well-demarcated capsular structure surrounding a hypoechoic core non-vascularized.

**Figure 5 diagnostics-13-01765-f005:**
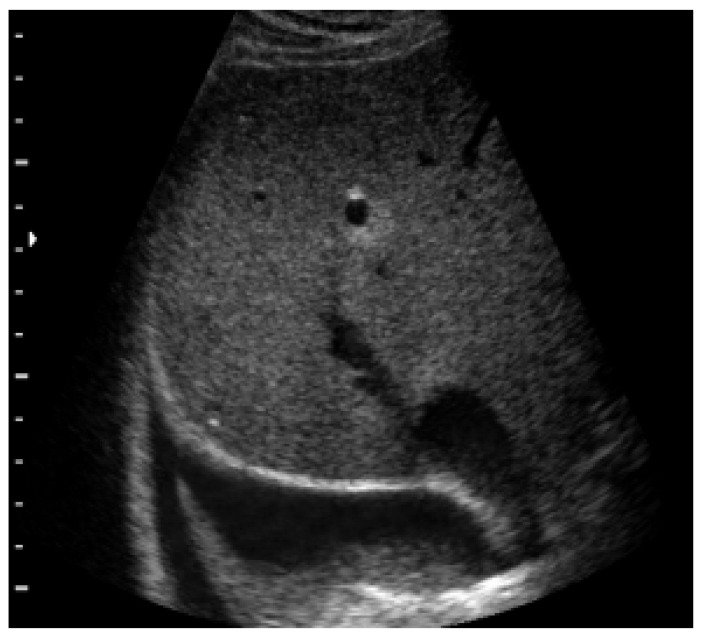
The figure shows a pleural effusion.

**Figure 6 diagnostics-13-01765-f006:**
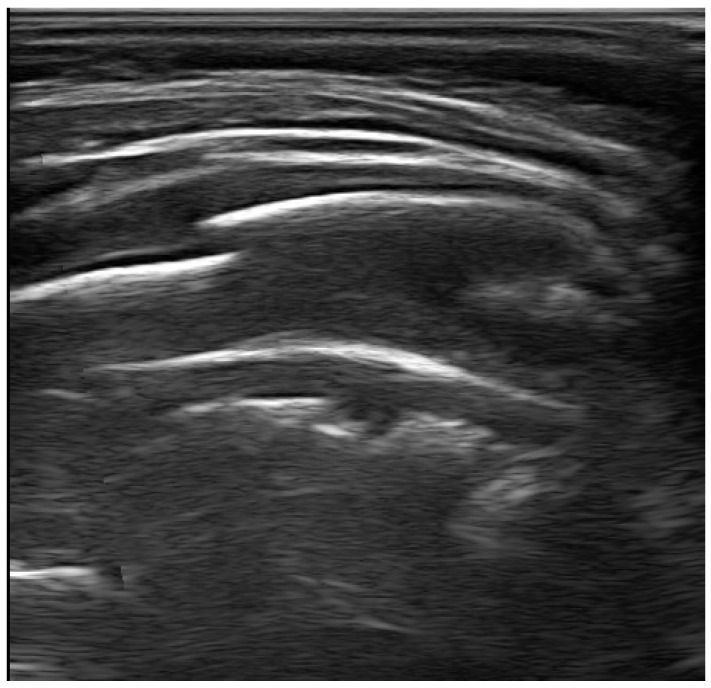
The figure shows a recent fracture of the seventh rib in transverse projection.

**Table 1 diagnostics-13-01765-t001:** Studies about the lung ultrasound iin bronchiolitis.

Study	Study Group	Study Type	Outcome	Key Results
La Regina DP et al. [8]	92 infants: Cases: 63 infants hospitalized for bronchiolitis. Controls: 29 infants hospitalized for non-respiratory diseases.	Comparative study	To correlate an LUS score with a clinical score. To describe lung ultrasound findings in cases and controls. To compare the LUS findings with chest X-ray (CXR) in infants hospitalized with bronchiolitis.	In infants with bronchiolitis, the LUS score showed a positive correlation with the clinical score and the length of hospitalization. LUS findings observed in the cases were the presence of B-lines, subpleural consolidations, and pleural line abnormalities. In patients who performed LUS and CXR, there was a correlation between the presence of abnormalities of the pleural line with the LUS and the presence of air trapping with CXR.
Supino MC et al. [10]	76 infants (from 1 to 12 months aged) with a clinical diagnosis of bronchiolitis.	Clinical trial	To evaluate if the ultrasound score can predict who among infants with bronchiolitis will need respiratory support and to evaluate the existence of a correlation between clinical presentation and US score.	Both clinical and ultrasound scores were able to predict the need for oxygen support in infants with bronchiolitis evaluated in the ED. The ultrasound score was found to correlate with the need for respiratory support (HFNC or HCPAP).
Bobillo-Perez S et al. [12]	80 infants (median age 53 days) with a clinical diagnosis of bronchiolitis.	Observational study	To evaluate the capacity of a new lung ultrasound score (LUSBRO) to predict the need for admission to the pediatric intensive care unit for mechanical ventilation, respiratory support duration, and the length of stay in the hospital.	The LUSBRO score is a useful tool to predict the need for admission to the PICU.
Bueno-Campana M et al. [13]	145 infants <6 months of age who were hospitalized for bronchiolitis.	Observational study	To predict the need for respiratory support in infants with bronchiolitis.	LU score ≥3.5 showed a sensitivity of 89.1% and specificity of 56% to identify patients at risk of needing respiratory support.
Taveira M et al. [14]	47 infants <6 months of age who were hospitalized for severe acute viral bronchiolitis.	Observational study	To evaluate the correlation between the severity of LU-diagnosed lung lesions using a quantitative LU score and the length of non-invasive ventilation (LOV).	LU score does not correlate with LOV.
Biagi C et al. [15]	87 children from birth to 24 months of age who were hospitalized for acute bronchiolitis.	Clinical trial	To assess the diagnostic accuracy and reliability of LUS for the detection of pneumonia in hospitalized children with bronchiolitis. To evaluate the agreement between LUS and CXR in diagnosing pneumonia in these patients.	There was a good accuracy of LUS in diagnosing bacterial pneumonia in children with clinical bronchiolitis. When including only consolidation size >1 cm, the specificity of LUS was higher than CXR.

**Table 2 diagnostics-13-01765-t002:** Studies about the role of lung ultrasound in pneumonia.

Study	Study Type	Outcome	Key Results
Stadler JAM et al. [19]	Review	To evaluate the evidence of diagnostic accuracy of the lung compared to other imaging modalities.	US examination can detect lung consolidation suggestive of pneumonia in children with similar accuracy and reliability as chest radiographs.
AM Pareda et al. [20]	Meta-analysis	To evaluate the accuracy of LUS for children with pneumonia.	LUS had a sensitivity of 96 and a specificity of 93% in the diagnosis of pneumonia.
DS Balk et al. [21]	Meta-analysis	To analyze the accuracy of lung ultrasound compared to CXR for the diagnosis of pediatric community-acquired pneumonia.	Lung ultrasound had significantly better sensitivity with similar specificity when compared to chest X-ray for the diagnosis of pediatric community-acquired pneumonia.
Bloise et al. [24]	Clinical trial	To determine the sensitivity and specificity of LUS in community-acquired pneumonia diagnosis compared with CXR.	Lung ultrasound showed a sensitivity of 97% and a specificity of 96% compared with CXR in the diagnosis of pneumonia.
G Iorio et al. [25]	Retrospective study	To analyze any differences between lung ultrasonography and chest radiography images in children with a diagnosis of community-acquired pneumonia (CAP).	Matched with CR, LUS detects more cases of pneumonia.

## Data Availability

Not applicable.

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
