# Peer review of "Point-of-Care Thoracic Ultrasound in Children: New Advances in Pediatric Emergency Setting"

_diagnostics, 2023, doi:10.3390/diagnostics13101765_

Round 1

Reviewer 1 Report (Previous Reviewer 1)

This article introduces the application of thoracic ultrasound in the field of children, and I wonder whether children include newborns. The very important defect of this paper is that the quality of the pictures is very poor, so it is recommended to replace or delete it.

Author Response

Dear Editors and Reviewers,

Thank you for your consideration of our manuscript. We truly think the manuscript is improved after the revisions suggested. Below we respond in detail to the minor revisions the reviewers raised. We now submit our revised manuscript for publication in Diagnostics.  We marked the revisions through the manuscript using the “track changes”.

Reviewer 1

This article introduces the application of thoracic ultrasound in the field of children, and I wonder whether children include newborns.

No, our manuscript considers the main applications of thoracic ultrasound in Pediatric Emergency departments. We do not consider applications in the neonatal period (intended up to 28 days of age) in this manuscript; in our department, newborns follow a different pathway of access to emergency care than pediatric patients.

The very important defect of this paper is that the quality of the pictures is very poor, so it is recommended to replace or delete it.

We thank the reviewer for his suggestion, we have now removed the low quality images and replaced some images with higher quality figures

Reviewer 2 Report (Previous Reviewer 2)

Authros updated revised manuscript. Overall, the revised manuscript is well updated so review paper is now well written. However, there are some remaining questions to be updated before publication.

1. Please correct Point-of-care to point-of-care in Line 22. Authors need to use lower case because the word is located in the center of the sentence. Please check in Line 58 too.

2. Please add ref.(Its low cost, availability,~) with ref. (https://journals.plos.org/plosone/article?id=10.1371/journal.pone.0249034).

3. A high frequency linear probe is normally higher than 15 MHz. Thus, authors need to correct 10 MHz to 15 MHz.

4. Authors can delete Acknowledgments section because authors do not specify any information.

5. Please provide the web link if it is possible in Figure 1 (BioRender.com) or provide the web link as a reference.

None

Author Response

Dear Editors and Reviewers,

Thank you for your consideration of our manuscript. We truly think the manuscript is improved after the revisions suggested. Below we respond in detail to the minor revisions the reviewers raised. We now submit our revised manuscript for publication in Diagnostics.  We marked the revisions through the manuscript using the “track changes”.

Reviewer 2

Authros updated revised manuscript. Overall, the revised manuscript is well updated so review paper is now well written. However, there are some remaining questions to be updated before publication.

We thank the reviewer for his comments.

  1. Please correct Point-of-care to point-of-care in Line 22. Authors need to use lower case because the word is located in the center of the sentence. Please check in Line 58 too.

Done

  1. Please add ref.(Its low cost, availability,~) with ref. (https://journals.plos.org/plosone/article?id=10.1371/journal.pone.0249034).

We have now added this new reference.

  1. A high frequency linear probe is normally higher than 15 MHz. Thus, authors need to correct 10 MHz to 15 MHz.

We have now corrected the frequency of liner probe.

  1. Authors can delete Acknowledgments section because authors do not specify any information.

We have now deleted Acknowledgments section

  1. Please provide the web link if it is possible in Figure 1 (BioRender.com) or provide the web link as a reference.

We have now added the web link in the legend of Figure 1.  

This manuscript is a resubmission of an earlier submission. The following is a list of the peer review reports and author responses from that submission.

Round 1

Reviewer 1 Report

As above.

Reviewer 2 Report

Authors submitted the revised manuscript without any reply letter of the previous reviewers so I checked the manuscript in the revised manuscript only. However, there are still critical remaining comments which need to be resolved.

1. The expression of "so much so that" need to be corrected. I still see there are many broken English grammar. Thus, authors must request your colleagues who are native English people.

2. For the readers who do not know point-of-care ultrasound, authors must define the word. Authors had better mention point-of-care ultrasound machine and then, the application of the machine is about the thoracic part,

3, There are still remaining references needed in the revised manuscript between line 103-105. Please check others very carefully.

4. This is review paper. However, there is no Table provided for previous research papers. Authors must summarize that in separate Table.

5. In the conclusion section, authors do not summarize the whole manuscript because authors showed several parts in the manuscript.

6. The motivation of the review paper is still questionable. Please clarify and emphasize that.

7. The description of Figure 1 need to be added.

8. For each part, authors need to provide some limitation of the technology or technical issues or problems related to image quality or senisitivity, or signal-to-noise ratio or other issues. Authors did not mention them from 1. Acute Heart diseases to 6. Foreign body identification.